# Integrative Physiological, Transcriptome, and Proteome Analyses Provide Insights into the Photosynthetic Changes in Maize in a Maize–Peanut Intercropping System

**DOI:** 10.3390/plants13010065

**Published:** 2023-12-24

**Authors:** Chao Ma, Yalan Feng, Jiangtao Wang, Bin Zheng, Xiaoxiao Wang, Nianyuan Jiao

**Affiliations:** 1College of Agriculture, Henan University of Science and Technology, Luoyang 471023, China; machao840508@163.com (C.M.); wangjiangtao0713@163.com (J.W.); z1072543428@126.com (B.Z.); xiaoxiaowang@haust.edu.cn (X.W.); 2College of Life Science, Wuchang University of Technology, Wuhan 430223, China; fengyalan2004@163.com

**Keywords:** maize, transcriptome, proteome, photosynthesis, maize–peanut intercropping

## Abstract

Intercropping is a traditional and sustainable planting method that can make rational use of natural resources such as light, temperature, fertilizer, water, and CO_2_. Due to its efficient resource utilization, intercropping, in particular, maize and legume intercropping, is widespread around the world. However, the molecular details of these pathways remain largely unknown. In this study, physiological, transcriptome, and proteome analyses were compared between maize monocropping and maize–peanut intercropping. The results show that an intercropping system enhanced the ability of carbon fixation and carboxylation of maize leaves. Apparent quantum yield (AQY), the light-saturated net photosynthetic rate (LSPn), the light saturation point (LSP), and the light compensation point (LCP) were increased by 11.6%, 9.4%, 8.9%, and 32.1% in the intercropping system, respectively; carboxylation efficiency (*CE*), the CO_2_ saturation point (Cisat), the Rubisco maximum carboxylation rate (Vcmax), the maximum electron transfer rate (Jmax), and the triose phosphate utilization rate (TPU) were increased by 28.5%, 7.3%, 18.7%, 29.2%, and 17.0%, respectively; meanwhile, the CO_2_ compensation point (Γ) decreased by 22.6%. Moreover, the transcriptome analysis confirmed the presence of 588 differentially expressed genes (DEGs), and the numbers of up-regulated and down-regulated genes were 383 and 205, respectively. The DEGs were primarily concerned with ribosomes, plant hormone signal transduction, and photosynthesis. Furthermore, 549 differentially expressed proteins (DEPs) were identified in the maize leaves in both the maize monocropping and maize–peanut intercropping systems. Bioinformatics analysis revealed that 186 DEPs were related to 37 specific KEGG pathways in each of the two treatment groups. Based on the physiological, transcriptome, and proteome analyses, it was demonstrated that the photosynthetic characteristics in maize leaves can be improved by maize–peanut intercropping. This may be related to PS I, PS II, cytochrome b6f complex, ATP synthase, and photosynthetic CO_2_ fixation, which is caused by the improved CO_2_ carboxylation efficiency. Our results provide a more in-depth understanding of the high yield and high-efficiency mechanism in maize and peanut intercropping.

## 1. Introduction

Light is one of the basic environmental factors for plant growth and development, which directly affects the photosynthetic characteristics of plants [1]. Crop yield is derived from the accumulation of photosynthetic products, which can be increased through improving the utilization efficiency of light energy and photosynthesis in the field [2]. This will be the direction of development for achieving high-yield and high-efficiency agriculture in the future. Intercropping is an intensive planting pattern in time and space, which can improve the utilization efficiency of light, temperature, water, fertilizer, the atmosphere, and other resources as well as the efficiency of light energy utilization and land productivity with a marginal effect [3,4,5]. It plays a critical role in agricultural production around the world. With the rapid reduction in land resources and the sharp growth in the world population, there is an urgent need to satisfy the global growth in food demand [6]. More and more countries will pay increased attention to intercropping and relay cropping. Maize is very important in many intercropping and interplanting systems [7,8,9]. Studies have shown that maize yield is significantly higher with intercropping than that with monocropping, such as maize–soybean [10], wheat–maize [11], and maize–peanut intercropping systems [12], revealing obvious advantages in terms of intercropping yield. In the intercropping system of leguminosae and gramineae, the underground parts of gramineae will compete to absorb nitrogen from the root area of leguminosae crops, thus reducing the “nitrogen repression” of leguminosae crops and promoting the symbiotic nitrogen fixation activity [13,14,15]. In the canopy part of an intercropping system, the distribution characteristics of the high pole and short pole changed the luminous environment of the entire crop canopy and significantly improved the rate of light interception [16]. Especially in calcareous soil, mugineic acid and other small-molecule organic acids secreted from maize roots could strongly chelate insoluble iron in the soil around crops in an intercropping system [17]. This has a crucial effect on iron nutrition absorption, especially in terms of improving the nitrogen fixation activity of leguminosae crops. Although there have been a large number of meaningful results regarding the study of complex ecosystems formed by intercropping, the related research should not be confined to physiological and ecological methods; the transcriptome and proteomics mechanisms should also be included.

Gene expression is a highly complex and intricate regulatory network that enables cellular metabolism to be in an orderly state in both time and space and facilitates a response to changes in environmental conditions [18]. Regulation of gene expression is the molecular basis for cell differentiation, morphogenesis, and individual development in living organisms. In transcription, a gene may produce a variety of mRNA alternative splicing and translate into different proteins, while the same protein may also undergo many forms of post-translational modifications to play various roles [19]. An increasing number of researchers are combining transcriptomics, proteomics, and metabolomics to explain scientific issues at the levels of species, genes, and metabolites through two or more omics research methods so as to better understand plant growth, development, and responses to stress. For example, in terms of plant growth and development, Chai et al. identified a gene encoding MADS-box gene *AGAMUS-like 42* (*AGL42*), which may be involved in the regulation of the flowering stage in *Brassica napus*, using transcriptome and proteome data [20]. Jia et al. revealed the molecular mechanisms underlying the responses of different rice varieties to low-temperature stress through a combination of transcriptome and proteome analyses. The genes and proteins involved in reproductive growth among rice varieties were differentially expressed under cold stress, and the translation regulation mechanism played an important role in the tolerance of rice to low-temperature stress [21]. Another interesting experiment using a combination of transcriptome and proteome analyses confirmed that grafting watermelon (*Citrullus lanatus*) as a scion onto a wax gourd (*Lagenaria siceraria*) can increase the salt tolerance of watermelon through changes in plant hormone signal transduction, amino acid synthesis, and photosynthesis [22]. Schmolinger et al. conducted a study on photosynthesis using *Chlamydomonas reinhardtii* as a model organism. Through a multi-omics analysis under nitrogen starvation conditions, it was confirmed that the transcripts and proteins that referred to the Calvin cycle were significantly reduced under nitrogen deficiency conditions. These reduced protein components mainly came from proteins with a high nitrogen content. This reduced cell nitrogen/carbon ratio may be a considerable mechanism for improving nitrogen utilization efficiency in organisms [23].

Maize–peanut intercropping is a typical planting pattern of legume crops intercropping with gramineous crops, playing an irreplaceable role in solving the contradiction of grain and oil competition for land [1]. However, the molecular basis of photosynthetic changes in maize leaves under maize and peanut intercropping remains unclear. In this research, RNA sequencing and the iTRAQ technique were employed to analyze transcriptome and proteomics responses in maize leaves between maize monocropping and maize–peanut intercropping at the silking stage of Zhengdan 958. This study not only contributes to revealing the molecular mechanism of maize yield advantages under intercropping, but also provide a theoretical basis for the efficient and high-yield research of intercropping and relay cropping.

## 2. Results

### 2.1. Photosynthetic Characteristic Changes in Response to Maize–Peanut Intercropping

The canopy characteristics of the high–low phase distribution between peanut and maize intercropping increased the maize ear leaf and canopy light quantum density by 20.4%, which obviously exceeded that of monoculture maize from 8 a.m. to 7 p.m. Compared with monoculture systems, the total biomass was increased by 36.1% in the intercropping system (Figure 1). Intercropping could significantly affect the photosynthetic light response curves, photosynthetic CO_2_ response curves, and their related parameters (Figure 2). The parameters of the light response curve, such as the apparent quantum yield (AQY), light-saturated net photosynthetic rate (LSPn), light saturation point (LSP), and light compensation point (LCP) were increased by 11.6%, 9.4%, 8.9%, and 32.1% in the intercropping system, respectively; the parameters of the photosynthetic CO_2_ response curves including carboxylation efficiency (CE), the CO_2_ saturation point (Cisat), the Rubisco maximum carboxylation rate (Vcmax), the maximum electron transfer rate (Jmax), and the triose phosphate utilization rate (TPU) were increased by 28.5%, 7.3%, 18.7%, 29.2%, and 17.0%, respectively, while the CO_2_ compensation point (Γ) decreased by 22.6% (Figure 3).

### 2.2. Primary Transcriptome and Proteome Data Analyses

The quality of the raw transcriptome and proteome data were first assessed. Principal component analysis (PCA) exhibited that distinctions between the two varieties and three site samples were evident. In regard to the transcriptome data, 72.16% of the differences between samples could be interpreted through PCA1 (84.72%) and PCA2 (12.09%) (Figure 4A). In regard to the proteome data, 99.05% of the variance between samples can be illustrated via PCA1 (88.36%) and PCA2 (10.69%) (Figure 4B). These results indicate that the differentially expressed genes (DEGs) and differentially expressed proteins (DEPs) showed a dynamic change pattern during the two treatments. The transcription and proteome data of the hierarchical clustering analysis (HCA) revealed low differences in biological replication. The correlation coefficient from the transcriptome and proteome data were up to 0.92 and 0.91 among the biological replicates, respectively. The PCA and HCA results indicated a good correlation between biological replicates. The above results proved the stability and repeatability of the results and provided a guarantee for the reliability of the results.

A total of 12.3 Gb of clean data was obtained after filtering, with a total of 22,976,739 to 26,424,079 reads for each library, as shown in Appendix A. The proportion of Q30 (sequences with a sequencing error rate of <0.1%) in all libraries exceeds 85%, while the GC content ranges from 56.46% to 58.39%. According to the results of the protein identification, the numbers of total spectra, peptide spectrum match (PSM), peptides, unique peptides, and protein groups were 291,823; 51,015; 22,092; 17,644; and 4997, respectively (Appendix A).

### 2.3. The mRNA and Protein Expression Changes in the Genome under Maize–Peanut Intercropping

According to the iTRAQ results, 82.9% of proteins detected in the transcriptome data accounted for 10.1% of the total transcripts detected (Figure 5A). A total of 588 genes showed a different expression in the leaves between the maize monocropping and maize–peanut intercropping systems. Of these, 383 were up-regulated and 205 were down-regulated (Figure 5B). In all samples including the two treatments, a total of >17,600 unique peptides (equivalent to 4997 proteins) were identified and tested in a comparative analysis. In this analysis, a single analytical experiment could only partially identify the pertinent peptides in the highly complicated mixture of peptides. Among the proteins detected by the ANOVA test, proteins that exhibited a less than 1.2-fold change and those with *p*-values > 0.05 were discarded. According to this standard, a total of 549 proteins were found to be differentially abundant during the two treatments. Therefore, we identified them as DEPs; during the comparisons of T2 and T1, 245 DEPs increased and 304 DEPs reduced, respectively. The overall data of the up- and down-regulated proteins can be seen in Figure 5C. In order to generally analyze the differences in gene expression at the transcriptional and translational levels, nine-quadrant association transcriptome and proteome analyses were carried out (Figure 5D). Quadrants 1, 2, and 4 indicate that the protein expression levels were lower than those for mRNA; while quadrants 6, 8, and 9 revealed the opposite trend. In quadrants 3 and 7, the protein expression levels were the same as those for mRNA, while quadrant 5 shows that there was no difference between the expression levels of protein and mRNA. According to the Pearson correlation coefficient (0.0516), it can be inferred that most differentially expressed proteins were not correlated with their corresponding RNA expression levels, indicating that this process may be regulated by post-transcriptional modifications. From the Venn diagram of differential expression between transcriptome and proteome analyses (Figure 5E), 62 RNAs and proteins were jointly up-regulated in the transcriptome and proteome, and 39 RNAs and proteins were collectively down-regulated in the transcriptome and proteome.

A search for DEGs in the GO database was carried out for enrichment analysis to assess the affected leaves during maize–peanut intercropping (Figure 6A). The molecular function (MF), cellular components (CC), and biological processes (BP) were evaluated through the GO database enrichment analysis. The enriched DEG terms focused on single-organism cellular processes (GO:0044763), mitochondrion (GO:0005739), and protein binding (GO:0005515) in BP, CC, and MF, respectively. A total of 331 DEPs were identified as uncharacterized proteins in Uniprot (http://www.uniprot.org/) (accessed on 1 December 2018). In addition, functional information was obtained by searching against the BLASTP (http://www.ncbi.nlm.nih.gov/BLAST/) (accessed on 1 December 2018). in the NCBI non-redundant (Nr) protein database. GO mapping and DEP annotation data are shown in Figure 6A. Through further enrichment analysis (http://bioinfo.cau.edu.cn/agriGO/) (accessed on 1 December 2018), it was found that differential proteins were enriched in biological processes (GO:0008150), cellular processes (GO:0009987), as well as metabolic processes (GO:0008152). In terms of the cellular component, DEPs were mainly detected in the cells (GO:0005623), cellular components (GO:0005575), cellular parts (GO:0044464), and intracellular parts (GO:GO:0044424). In the biological process, DEPs were mostly involved in binding (GO:0005488), catalytic activity (GO:0003824), and molecular function (GO:0003674).

The GO analysis revealed that DEGs, which were found to be generalized throughout the maize leaves in response to intercropping, were affected by different cellular processes, particularly, metabolic processes (Figure 6B). In the transcriptome data, DEGs mainly focused on ribosomes (path: ko03010), plant hormone signal transduction (path: ko04075), and photosynthesis (path: ko00195). In terms of the proteome data, of the 549 DEPs, 186 were related to 37 specific KEGG pathways during the paired comparison of T2/T1. In addition, 84 up-regulated proteins were further analyzed to investigate the function of the potential metabolic pathway among the paired comparisons mentioned above. We found that these proteins mainly participate in pathways involving spliceosomes (path: ko03040), carbon metabolism (path: ko01200), phenylpropanoid biosynthesis (path: ko00940), viral carcinogenesis (path: ko05203), and oxidative phosphorylation (path: ko00190). It is worth noting that DEPs were also related to photosynthesis (ko00195). In this proteomics study, the photochemical reaction center (PS I, PS II, the cytochrome b6f complex, and ATP synthase) and photosynthetic CO_2_ fixation were focused. The functions of these screened proteins are discussed below.

### 2.4. Photosynthetic Response Transcriptome Differences under Maize–Peanut Intercropping

A total of 588 DEGs were used to construct a hierarchical clustering analysis (Figure 7). The processes of PS I, PS II, the cytochrome b6f complex, ATP synthesis, the light-harvesting chlorophyll protein complex, and CO_2_ fixation have been focused on, as well as genes involved in the C3 and C4 photosynthetic pathways. In PS I, Psa C gene expression was significantly reduced. In PS II, the expression of the D1 and Psb 28 genes was dramatically reduced, while the expression of the Psb Y gene noticeably increased. In ATP synthesis, the delta chain gene was notably down-regulated. The expression levels of Lhcb 1 and Lhcb 3 were remarkably reduced in the light-harvesting chlorophyll protein complex. Interestingly, during the CO_2_ fixation process, the expression level of phosphoenolpyruvate carboxylase increased by 1.9 times. These results suggested that noteworthy changes in the light environment of the maize canopy under intercropping conditions triggered expression alterations for photosynthetic-related genes (Figure 7).

### 2.5. Photosynthetic Response Proteome Differences under Maize–Peanut Intercropping

Plants produce organic matter through photosynthesis, which determines the yield of crops. To further elucidate the molecular mechanism of photosynthetic advantages under maize–peanut intercropping conditions, the expressions of photosynthetic-related proteins were mainly detected, including PS I, PS II, the cytochrome b6f complex, ATP synthase, and CO_2_ fixation (Figure 8). Compared with a monocrop of maize, the PS I-LHC I subunit Psa V increased by 34%. Although its specific function in photosynthesis has not been reported, it is speculated that it might play a key role in absorption, transformation, and the transformation of light energy. The expression of cytochrome Cyt b559 β subunit increased by 28% in intercropped maize leaves, which is encoded by psbF and plays a central role in PS II. Psb L, chlorophyll a/b binding protein 1 (CAB1), and chlorophyll a/b binding protein 2 (CAB2) decreased by 57%, 21%, and 35%, respectively. This may be due to the improved light environment in the maize canopy under intercropping conditions. The expression of blue copper protein (BCP), which is involved in the regulation of biological electron transfer, exhibited a significant 2.02-fold up-regulation under the intercropping system, indicating that the photosynthetic electron transport efficiency of maize leaves was enhanced under intercropping conditions. ATPsB expression increased by 31%, which is involved in photophosphorylation. The photoactivated phosphoenolpyruvate carboxylase (PEPC), pyruvate orthophosphate dikinase (PPDK), and malate dehydrogenase (MDH) increased by 64.7%, 33%, and 26.5% throughout the C4 cycle, respectively. Meanwhile, in the Calvin cycle, the Rubisco large subunit-binding protein subunit α (rbcLBP), Rubisco accumulation factor 1 (RAF1), CP12-1 protein (CP12-1), and malic enzyme (ME) rose by 23%, 22%, 20%, and 26%, respectively. In addition, Clp protease (Clp P), which is involved in chloroplast development, declined by 57%. The superoxide dismutase (SOD), peroxidase (POD), and AP2-EREBP transcription factors (AP2) associated with oxidative stress and aging grew by 26%, 92%, and 79%, respectively. These results confirm the results of our photosynthetic physiological experiments (Figure 1, Figure 2 and Figure 3).

### 2.6. Transcriptional Expression Analysis by qRT-PCR

In order to further confirm the difference in the transcription expression patterns, a relative expression Analysis of 10 genes was carried out by qRT-PCR (Figure 9). The transcription levels of nine genes showed a similar tendency with their corresponding proteins, such as CAB1, BCP, PEPC, PPDK, MDH, CP12-1, ME, SOD, and POD. However, only one gene (rbcLBP) showed the different expression pattern with its corresponding protein. Such differences might be explained by the complex translation regulation mechanism in the intercropping system.

### 2.7. String-Based Protein–Protein Interaction (PPI) Analysis

The string database (https://string-db.org/) (accessed on 25 September 2023) is a search database for protein–protein interactions, including both direct physical interactions and indirect functional correlations. A protein–protein interaction network was constructed using 549 DEPs (Figure 10). The interaction network consists of 47 nodes (47 proteins). The previous results of the KEGG enrichment analysis showed that the transcriptome of DEGs was significantly enriched in the photosynthesis (path: ko00195) pathways. What is more, DEPs were also involved in the pathway of photosynthesis (Figure 8). In the protein–protein interaction (PPI) network, two DEPs (red circle) encode ferredoxin-2 and photosystem I reaction center subunit V. Iron redox protein is an iron sulfur protein that transfers electrons in various metabolic reactions. It plays a crucial role in transferring the photoreduction ability to Fd NADP^+^ oxidoreductase, thereby forming NADPH and mediating the circulating electron flow around photosystem I. Psa V is an important photosystem I reaction center subunit. These results suggest that FDX2 and Psa V may play a role in the photosynthetic changes in the maize kernel in a maize–peanut intercropping system.

## 3. Discussion

Gramineae–leguminosae intercropping is a common planting system in East Asia and Africa [24]. In recent years, it has developed rapidly in Sichuan, Gansu, Guangdong, and the Huang-Huai-Hai region in China. The canopy structure in terms of the high-stalk and short-stalk arrangement not only changes the light-receiving state of the single population in the later growth stage of maize, but also improves the light-receiving conditions of the middle and lower leaves [25]. Moreover, the occurrence of premature senescence is also significantly alleviated [26]. Maize is a typical C4 crop with a higher light saturation point. The improvement of light conditions significantly enhances the ability of maize leaves to utilize light, and the improved photosynthetic characteristics are finally reflected in the yield advantage of intercropping [27]. The present study showed that maize–peanut intercropping can improve the canopy light distribution and the light energy utilization. LCP and LSP are the direct responses in plant leaves to the external light environment, which reflect the adaptation status of plant leaves in low-light and high-light conditions [28]. In this study, the intercropping system significantly increased the AQY, LSPn, LSP, and LCP in maize ear leaves and enhanced the utilization efficiency of light in maize ear leaves. CE is sensitive to changes in light and CO_2_ concentrations and is also a limiting factor for photosynthetic carbon assimilation [29]. Intercropping has significantly increased the Cisat, CE, Vcmax, Jmax, and TPU in maize ear leaves, which indicates that the farming methods enhanced the ability of carbon fixation and the carboxylation of the ear leaves. However, to a great extent, the output advantage of intercropping is directly related to the interaction between the above- and below-ground intercropping crops. A large number of studies have aimed to determine the responses of maize in an intercropping system in terms of physiology and ecology. However, the molecular mechanism of the advantage in terms of the maize yield needs further study.

Protein is not only the structural substance of organisms, but also the catalyst of many biochemical reactions [30]. In spite of transcriptomics being useful in revealing the impact on intercropping, the ultimate expression of life activity is the protein. Therefore, proteomics is beneficial to understanding the complex regulatory mechanism of an intercropping system on maize growth [12]. To further elucidate the molecular mechanism of maize–peanut intercropping, transcriptome analysis and proteome quantitative analysis based on the iTRAQ technique were performed in maize monocropping and maize–peanut intercropping systems. The results showed that 588 DEGs and 549 DEPs were identified in a paired comparison of T2/T1. Transcriptome and proteome data demonstrated that the DEGs and DEPs were involved in vital yield formation metabolism pathways, which may play a significant role in the maize yield advantage under an intercropping system. In addition, translation regulation and post-translational regulatory mechanisms may exert an influence on the photosynthetic changes in maize leaves under maize–peanut intercropping. The five photosynthetic functional categories were considered in the evaluation. These metabolic pathways included PS I, PS II, the cytochrome b6f complex, ATP synthase, photosynthetic CO_2_ fixation, and reactive oxygen species (ROS) for discussion.

The photoreaction process of photosynthesis mainly involves four protein complexes, which are present on the thylakoid membrane, including PS I, PS II, the cytochrome b6f complex, and the ATP synthase complex. The first three complexes constitute the photosynthetic electron transport system on the thylakoid membrane, i.e., the electron transport chain or photosynthetic chain [31]. The structure and function of PS I have always been the focus and hotspot of photosynthesis research. The size of PS I is quite small, and it is outside the thylakoid membrane and composed of a reaction center and a light-harvesting pigment complex (LHC) [32]. The PS I–LHC I exist in plants in the monomeric form, with each monomer consisting of at least 14 core subunits (Psa A~L, Psa N, Psa O) [33]. There are 10 subunits (Psa A~F, Psa I~L) that are conserved in plants and cyanobacteria, and the remaining four subunits (Psa G~H, Psa N~O) are specific to higher plants, while Psa M and Psa X are unique to cyanobacteria [33]. In this experiment, we first identified Psa V in the PS I of maize, and the expression of this subunit was dramatically up-regulated in the intercropping system. Steppuhn J. et al. identified *Psa V* in spinach, whose cDNA was 677 bp, the molecular weight of the precursor protein was 18.2 kDa, and the molecular weight of the mature polypeptide was 10.8 kDa. However, the function of Psa V has not been studied so far [34].

Compared with PS I, the size of PS II is larger, and it is located inside the thylakoid membrane [32]. It is mainly composed of three parts: the reaction center complex, LHC, and the oxygen-evolving complex (OEC) [35]. The PS II protein complex contains the inner antenna protein CP, as well as two core polypeptides, D1 and D2, which are all essential components of the PS II complex [32]. P680, Pheophyll (Pheo), and special plastoquinone (QA and QB) are combined on D1 and D2 [36]. The outer layer of PS II is the PS II concentrated light-harvesting pigment complex (LHCII), which is composed of a photosynthetic pigment and a protein [37]. Compared with photosynthetic bacteria, higher-plant PS II contains a unique cytochrome, Cyt b559, consisting of α and β subunits, encoded by psbE and psbF, respectively [37,38]. Previous studies have revealed that Cyt b559 has a central effect on the function of PS II [38]. The intercropping in this study remarkably increased the expression of Psb F, which also supports the results that the intercropping system could markedly improve the photosynthetic performance of maize. In addition, CAB1 and CAB2 are involved in the initial reaction process of photosynthesis, both of which involve the absorption, transfer, and conversion of light energy in photosynthetic pigments [39]. The expressions of Psb L, CAB1, and CAB2 were down-regulated since the light conditions of the leaves of the maize canopy were significantly improved in the intercropping system.

Electron transfer from PS II to PS I is mediated by Cyt b6f, which is regulated by BCP [40]. In this study, the expression of BCP significantly increased by 2.02-fold in the intercropping system. The photosynthetic electron transport efficiency of maize leaves was also enhanced under intercropping conditions. ATPsB, involved in photophosphorylation, rose by 35% under intercropping conditions [39]. This also reasonably explained the results of our physiological data: intercropping could promote the photosynthetic efficiency of maize leaves.

Maize originating from the tropics has two pathways for fixed CO_2_: the C4 and Calvin cycles [5,12]. In the C4 pathway, the receptor for CO_2_ is phosphoenolpyruvate (PEP), and the fixation of CO_2_ occurs in the cytoplasm of mesophyll cells. PEPC catalyzes the reaction of PEP with CO_2_ to form some C4 acids, such as oxaloacetate and malic acid; these are decarboxylated after they are transferred to bundle sheath cells, and the Rubisco fixes the released carbon dioxide again [40,41]. The mesophyll cells of the maize parenchyma contain sheath cells, and they can form a Kranz structure that acts as a CO_2_ pump. This effectively assimilates the external CO_2_ and transports CO_2_ in the form of C4 acid to the periphery of Rubisco, maintaining a high carbon assimilation efficiency [42]. Thus, the up-regulation of the PEPC protein may be the main reason for the assimilation ability of maize leaves, which is greatly promoted by an intercropping system. In this study, the expressions of PEPC, PPDK, and MDH in maize leaves increased by 64.7%, 33%, and 26.5%, respectively. Previous studies have shown that light can activate PEPC, PPDK, and MDH, which is caused by changes in the canopy illumination of maize leaves [43].

In the Calvin cycle, the receptor for CO_2_ is RuBP, and fixation of CO_2_ occurs in vascular sheath cells [29]. The Clp P in higher plants has an important diverse proteolytic core function. This exerts an essential regulatory effect on the development and functional maintenance of chloroplasts, indicating that Clp P is quite crucial in the C4 pathway [44]. Under intercropping conditions, the expressions of rbcLBP, RAF1, CP12-1, and malic enzyme (ME) increased by 23%, 22%, 20%, and 26%, respectively. The most important enzyme in the Calvin cycle, Rubisco, is composed of several large and small subunits [44]. Studies have shown that the large subunits have catalytic functions, and the small subunits only have regulatory effects [45]. CP12-1 could regulate the content of phosphoricbulokinase (PRK) in the leaves and maintain the photosynthetic capacity of the leaves, while RAF1 can form a complex with rbcL to jointly regulate the assembly of Rubisco [46,47]. Furthermore, ME exerts a critical role in plant photosynthesis and the TCA cycle [29]. These results indicate that the photosynthetic CO_2_ fixation characteristics of maize leaves were noticeably improved under intercropping conditions.

ROS can be produced in many life processes of plants, especially photosynthesis. Since the stay-green ability is one of the important characteristics of a high yield, the ROS or photooxidation produced by photosynthesis or excess light would destroy the chloroplast structure, damage the function of photosynthetic organs, and cause premature aging, ultimately leading to a reduced yield [48]. The intercropping system significantly affected the distribution of light at different levels in the maize canopy, especially in the leaves in the lower middle part of the canopy. The improved ventilation and light conditions dramatically increased the photosynthetic performance of maize leaves, especially the ear leaves located in the middle of the canopy. This greatly alleviated the premature senescence in the lower middle leaves [25]. In this experiment, the expressions of SOD and POD increased by 26% and 92%, respectively, and the aging-related transcription factor AP2-EREBP rose by 79%. This indicated that intercropping could enhance the antioxidant capacity and anti-aging ability of maize leaves and was conducive to the performance of photosynthesis [49]. Moreover, the chlorophyll a/b binding protein (CAB) that was down-regulated in maize leaves under the intercropping system can alleviate the light damage caused by the ROS; ROS were generated by the excessive energy state on the thylakoid membrane [50]. Therefore, the changes in these photosynthetic-related metabolic pathways are beneficial to understanding the molecular mechanisms concerning the regulation of maize yield in an intercropping system (Figure 11).

## 4. Materials and Methods

### 4.1. Plant Growth Condition, Treatments, and Sampling

The experiment was carried out from May to September 2018, in the experimental field of Henan University of Science and Technology, located at 34°41′ N and 112°27′ E, at an altitude of 280 m above sea level. The mean annual rainfall and temperature in the region are 578.2 mm and 14.8 °C, respectively. The soil of the experimental farm is fluvo-aquic soil and had the following physicochemical characteristics at the beginning of the study: pH (H_2_O) = 7.08, available N = 80.09 mg·kg^−1^, available P = 3.31 mg·kg^−1^, available K = 81.32 mg·kg^−1^, and organic matter = 14.5 g·kg^−1^. The experiment involved the cultivar of maize ‘ZD 958’ in two planting systems: monocultured maize (T1) and maize intercropped with peanut ‘HY 16’ (T2). The row spacing and plant spacing of the monoculture corn were 60 cm and 25 cm, while those of intercropping system were 40 cm and 25 cm, respectively. The row spacing and plant spacing of intercropped peanuts were 30 cm and 20 cm, respectively. The intercropped system combined 3 sets of 4 rows of peanut plants grown in 120 cm wide strips. This was alternated with two rows of maize grown in 80 cm narrow strips, so the total width of the two crop strips was 200 cm. Each experimental plot was 60 m^2^ (6 m × 10 m), repeated 4 times. The two crops were sown on 20 May and harvested on 25 September. A total of 90 kg·hm^−2^ of N was applied as the base fertilizer, and another 90 kg·hm^−2^ of N was applied as a dressing fertilizer in the flare opening of maize only. For each sample, the middle of the ear leaf was harvested at the silking stage. The samples were immediately frozen in liquid nitrogen for 10 min and were then conserved in dry ice and mailed to the sequencing company as soon as possible. All physiological, transcriptomics, and proteomics samples included three independent biological replicates. The growth state is displayed in Figure 12.

### 4.2. RNA Extraction, Library Construction, and Sequencing

The total RNA of the samples was extracted according to the TaKaRa MiniBEST plant RNA extraction kit (TaKaRa, Dalian, China), and the purity, concentration, and integrity of RNAs were tested before constructing a cDNA library. The cDNA libraries were obtained through PCR amplification and sequenced using the Illumina platform from Biomarker Technology Co., Ltd. (Beijing, China). Sequence alignment was performed with the reference genome of *Zea mays* (Zm-B73-REFERENCE-NAM_5.0. new), and the unique mapped genes were obtained using HISAT2 (http://daehwankimlab.github.io/hisat2/) (accessed on 23 May 2020) and StringTie 2.2.0 software [51,52]. Multiple databases were utilized to perform functional annotation on individual genes.

### 4.3. Protein Extraction, Concentration Determination, Enzymatic Digestion, and iTRAQ Reagent Labeling

The protein from the leaf samples was extracted using acetone (containing 10% TCA, 65 mM DTT) as the extracting solution. Concentrations of the protein sample were measured using the BCA protein assay kit (Sangon, Shanghai, China). Digestion and labeling were conducted using the iTRAQ Reducing Reagent and Cysteine-Blocking Reagent (AB Sciex, CA, USA).

### 4.4. Chromatography, LC–MS/MS Analysis, and Raw Data Analysis

The detailed processes of chromatography and LC–MS/MS were conducted following Wang’s method [22]. A Q-Exactive mass spectrometer (Thermo, MA, USA) was used in the separation according to Uniprot_Maize_87279_20150928.Fasta, using Proteome Discoverer 1.4 software to analyze the raw data (http://www.uniprot.org/, downloaded 28 September 2015). The identified peptides were matched with proteins using Wang’s method [22]. The ratio between the 2 treatments was used to calculate the fold changes in T2 vs. T1. The method of protein classification was conducted following Wu’s method [53]. KEGG annotations were analyzed using KOBAS 2.0 software to identify the involved pathways [54].

### 4.5. Total RNA Extraction and qPCR

Total RNAs were extracted using a TRNzol Kit (CWBIO, Taizhou, China), and the cDNAs were obtained using a cDNA synthesis kit (CWBIO, Taizhou, China). Relative quantification was performed via a LightCycler^®^ 96 (Roche, Basel, Switzerland) instrument with a SYBR Green mix reagent (CWBIO, Taizhou, China). Designed primers and the selected *ZmGAPDH* gene were used as the internal reference (Appendix A).

### 4.6. Assay of Photo Flux Density and Photosynthetic Characteristics

Photo flux density was measured at the height of ear leaf of maize using the SUNSCAN instrument (Delta-T, Cambridge, UK). Photosynthetic parameters were measured on sunny days from 9:30 a.m. to 11:30 a.m., 10 days after the silking stage, using a portable photosynthetic analyzer (Li6400, Li COR, Lincoln, USA). The same measurement site as for the transcriptome and proteome analyses was used. The method for measuring the photosynthetic light response curve was according to Ma’s report [55]. The determination of the photosynthetic light intensity response curve included setting the light intensity gradients to 2000, 1800, 1500, 1200, 1000, 800, 600, 400, 300, 200, 150, 100, 50, and 0 μmol·m^−2^·s^−1^. Instrument parameter settings: ambient CO_2_ concentration, 400 μmol·mol^−1^; leaf chamber temperature, 30 °C; leaf chamber relative humidity, 50–55%. The photosynthetic CO_2_ response curves were measured in terms of the method by Long and Bernacchi (2003) [56]. Based on the measured light response curve, the saturation light intensity was determined, and a fixed light intensity was set. After sufficient induction, Pn was measured at CO_2_ concentrations of 400, 350, 300, 250, 200, 150, 100, and 50 μmol·mol^−1^; then, the CO_2_ concentration was adjusted back to 400 μmol·mol^−1^. When Pn was stable, the CO_2_ concentration was increased by 100~200 μmol·mol^−1^ each time until Pn no longer increased with the increase in CO_2_ concentration. The response curve data and Photosyn assistant software 1.0 were used to analyze and calculate the LCP, LSP, LSPn, AQY, CE, Γ, Cisat, Vcmax, Jmax, and TPU.

### 4.7. Statistical Analysis

All data are shown as the mean ± standard deviation (SD). The data were processed using Microsoft Excel 2016 and SPSS 17.0 statistical software, and the LSD method was adopted for intergroup comparison.

## 5. Conclusions

The intercropping system enhanced the ability of carbon fixation and the carboxylation of maize leaves. Transcriptome and proteome analyses of maize leaves under an intercropping system were carried out based on Illumina and iTRAQ technology to further understand the molecular mechanism behind the improvement in photosynthetic performance. A total of 588 DEGs and 549 DEPs were identified and quantified in the comparison of treatment combinations, T2/T1, indicating that the multiplexed gene expression patterns were complicated. The results of a large-scale comparison of DEPs in monocultured maize and maize–peanut intercropping revealed that the DEPs were involved in PS I, PS II, the cytochrome b6f complex, ATP synthase, and photosynthetic CO_2_-fixation-related metabolic pathways, and these may respond to an intercropping system.

## Figures and Tables

**Figure 1 plants-13-00065-f001:**
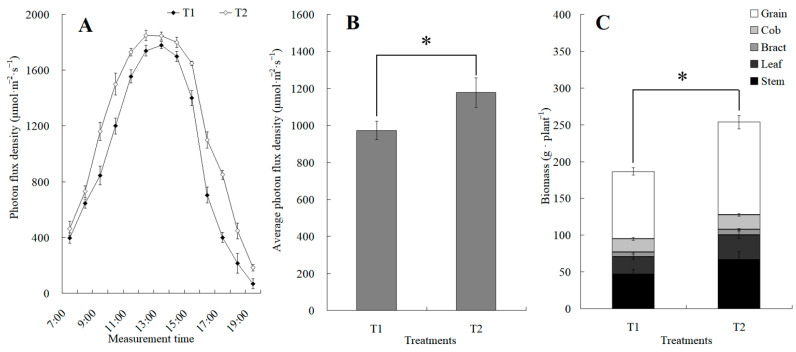
The changes in diurnal photon flux density and its average value between the maize monocropping and maize–peanut intercropping. (**A**) Determination of photon flux density at the ear leaf height of maize, which was conducted hourly from 7:00 to 19:00. (**B**) Average value of photon flux density. (**C**) Biomass of maize. * denotes a significant difference (*p* < 0.05). T1, maize monocropping; T2, maize–peanut intercropping.

**Figure 2 plants-13-00065-f002:**
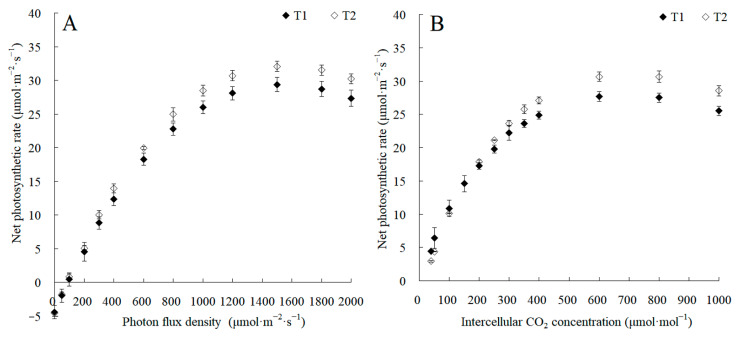
The changes in photosynthetic light response curves and CO_2_ response curves between the maize monocropping and maize–peanut intercropping systems. (**A**) Photosynthetic light response curves were recorded at 2000, 1800, 1500, 1200, 1000, 800, 600, 400, 300, 200, 100, 50, and 0 μmol·m^−2^·s^−1^ of photon flux density at 9:00–11:00. (**B**) Photosynthetic CO_2_ response curves were recorded at 40, 50, 100, 150, 200, 250, 300, 350, 400, 600, 800, and 1000 μmol·mol^−1^ of intercellular CO_2_ concentrations at 9:00–11:00. T1, maize monocropping; T2, maize–peanut intercropping.

**Figure 3 plants-13-00065-f003:**
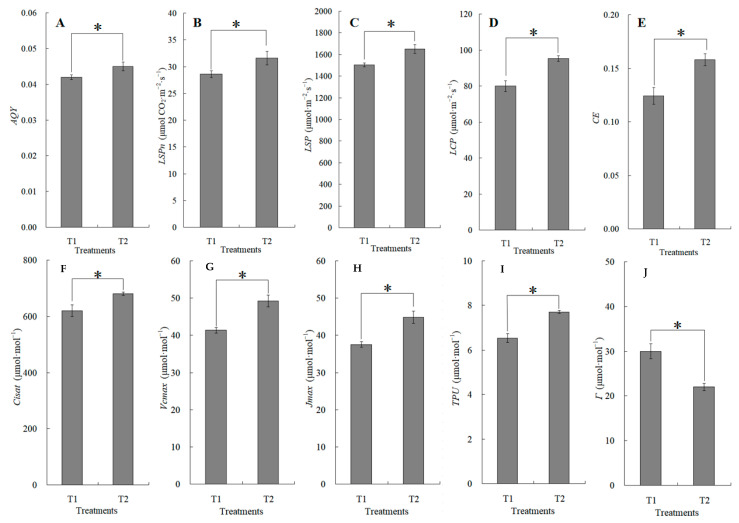
The changes in apparent quantum yield (AQY, (**A**)), light-saturated net photosynthetic rate (LSPn, (**B**)), light saturation point (LSP, (**C**)), light compensation point (LCP, (**D**)), carboxylation efficiency (CE, (**E**)), CO_2_ saturation point (Cisat, (**F**)), Rubisco maximum carboxylation rate (Vcmax, (**G**)), maximum electron transfer rate (Jmax, (**H**)), triose phosphate utilization rate (TPU, (**I**)), and CO_2_ compensation point (Γ, (**J**)) between the maize monocropping and maize–peanut intercropping systems. AQY, LSPn, LSP, and LCP were calculated by the Photosyn Assistant software 1.0, fitted by photosynthetic light response curves (Figure 2A). CE, Cisat, Vcmax, Jmax, TPU, and Γ were calculated by the Photosyn Assistant software 1.0, fitted by CO_2_ response curves (Figure 2B). * denotes a significant difference (*p* < 0.05). T1, maize monocropping; T2, maize–peanut intercropping.

**Figure 4 plants-13-00065-f004:**
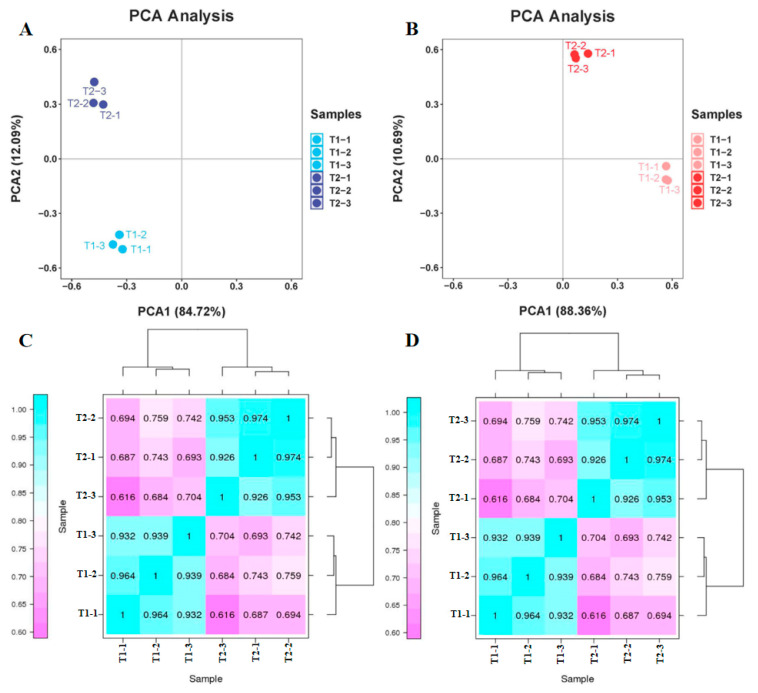
Principal component analysis (PCA) and hierarchical cluster analysis (HCA) from transcriptome and proteome data. (**A**) PCA of transcriptome data. (**B**) PCA of proteome data. (**C**) HCA of transcriptome data. (**D**) HCA of proteome data.

**Figure 5 plants-13-00065-f005:**
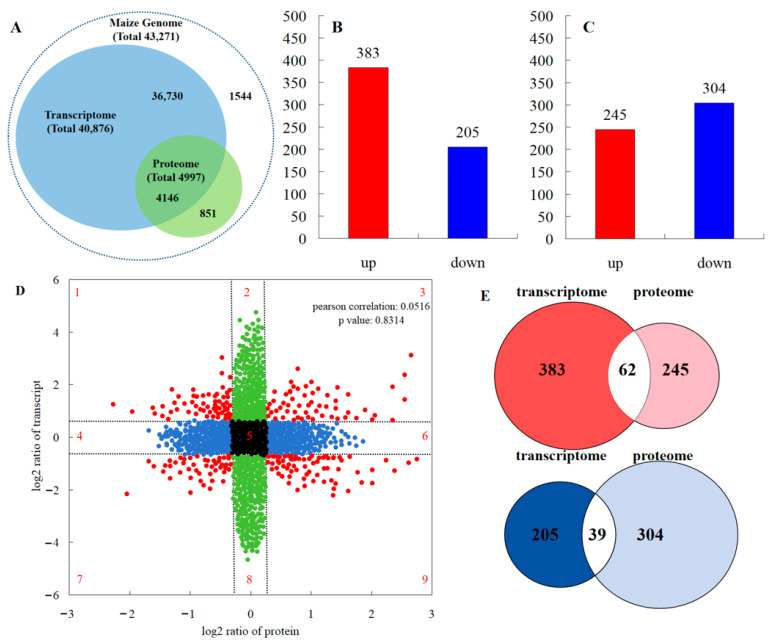
Comparisons of the protein and transcript abundances in the leaves between maize monocropping and maize–peanut intercropping systems. (**A**) Congruency between the detected transcripts and the proteins of the maize leaves. (**B**) Number of DEGs (FC ≥ 1) with FDR ≤ 0.01). (**C**) Number of DEPs (absolute value of log2 (1.2-fold change with *p* value < 0.05). (**D**) Nine quadrant association transcriptome and proteome analyses. The dashed lines divide the graph into nine parts. The black dots indicate that neither the transcripts nor the proteins are differentially expressed; the blue dots indicate that proteins are differentially expressed but transcripts are not; the green dots indicate that transcripts are differentially expressed but proteins are not; the red dots represent both differentially expressed transcripts and proteins. (**E**) Venn diagram of transcriptome and proteome differential expression. Red represents up-regulation; blue represents down-regulation.

**Figure 6 plants-13-00065-f006:**
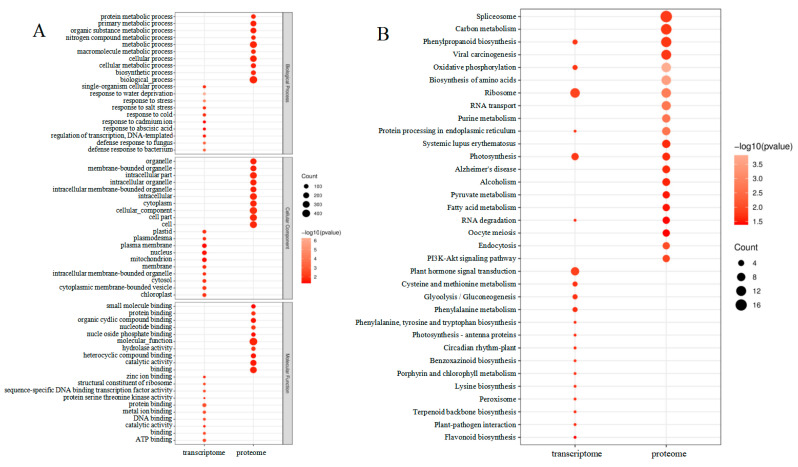
Enrichment analysis of transcriptome and proteome changes between maize monocropping and maize–peanut intercropping. (**A**) GO enrichment analysis of the differentially expressed genes and proteins. (**B**) KEGG enrichment analysis of the differentially expressed genes and proteins.

**Figure 7 plants-13-00065-f007:**
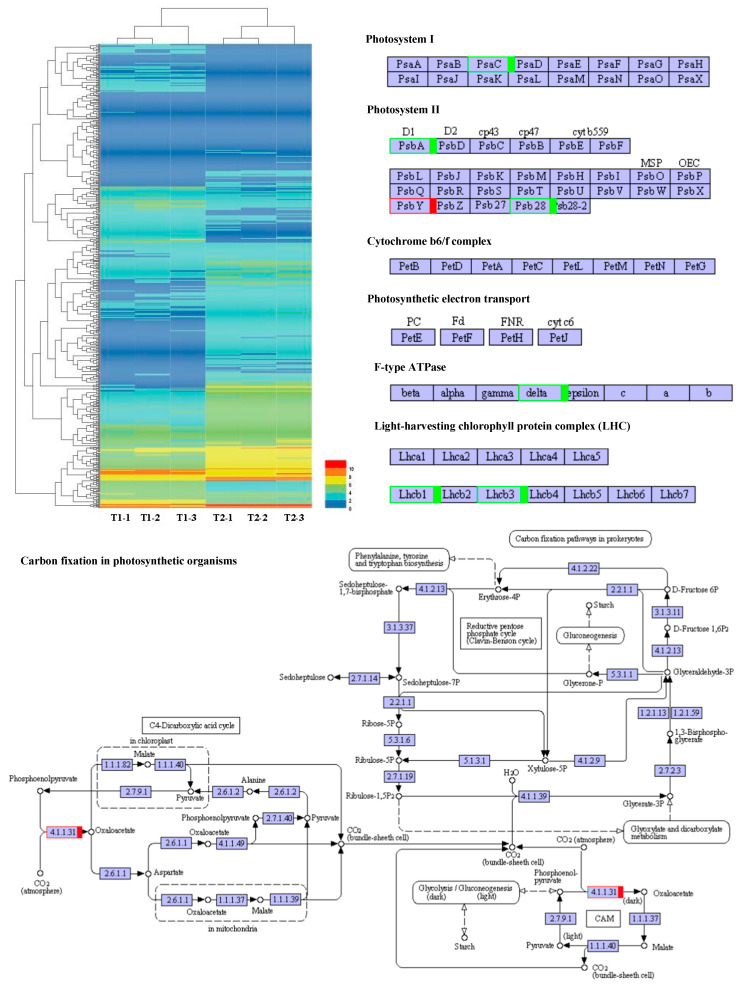
Photosynthetic response transcriptome differences under maize–peanut intercropping. T1, maize monocropping; T2, maize–peanut intercropping. Red represents up-regulated expression transcripts, green represents down-regulated expression transcripts.

**Figure 8 plants-13-00065-f008:**
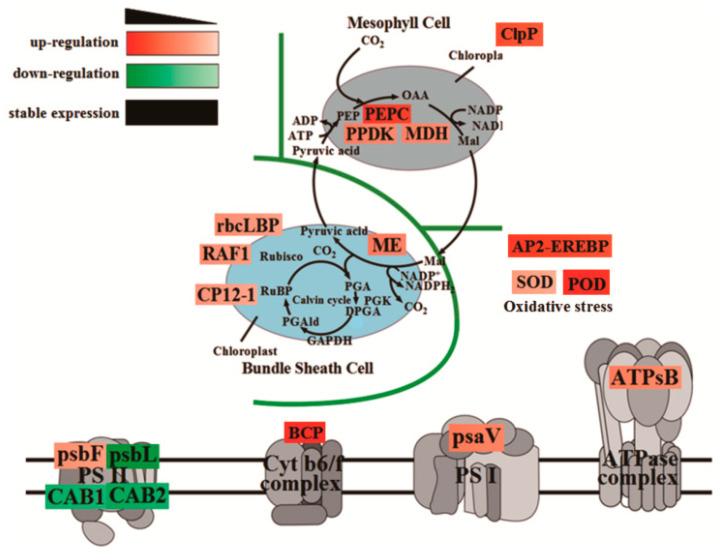
The variations in the abundance of DEPs involved in photosynthesis efficiency in maize leaves between the maize monocropping and maize–peanut intercropping systems. The DEP abundances are denoted in colorful rectangular grids. Red represents up-regulation, green represents down-regulation, and black words without rectangular boxes indicate stably expressed proteins. Gradation of color represents the degree of the changes.

**Figure 9 plants-13-00065-f009:**
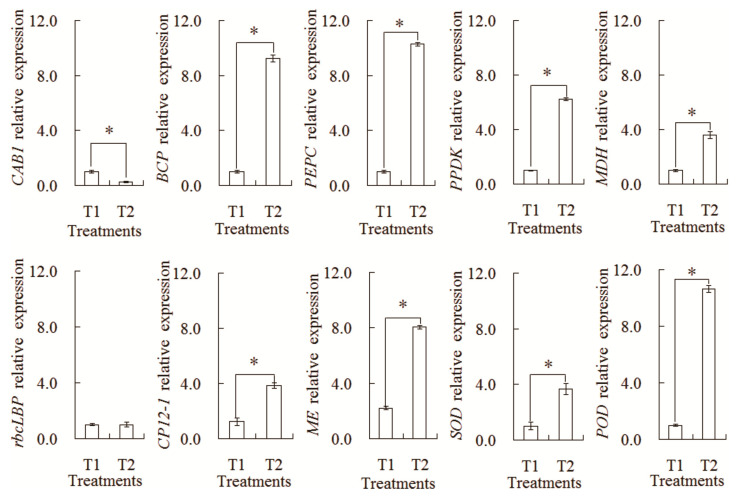
Analysis of transcript levels of the differential abundances of protein species between the maize monocropping and maize–peanut intercropping systems. * denotes a significant difference (*p* < 0.05). T1, maize monocropping; T2, maize–peanut intercropping.

**Figure 10 plants-13-00065-f010:**
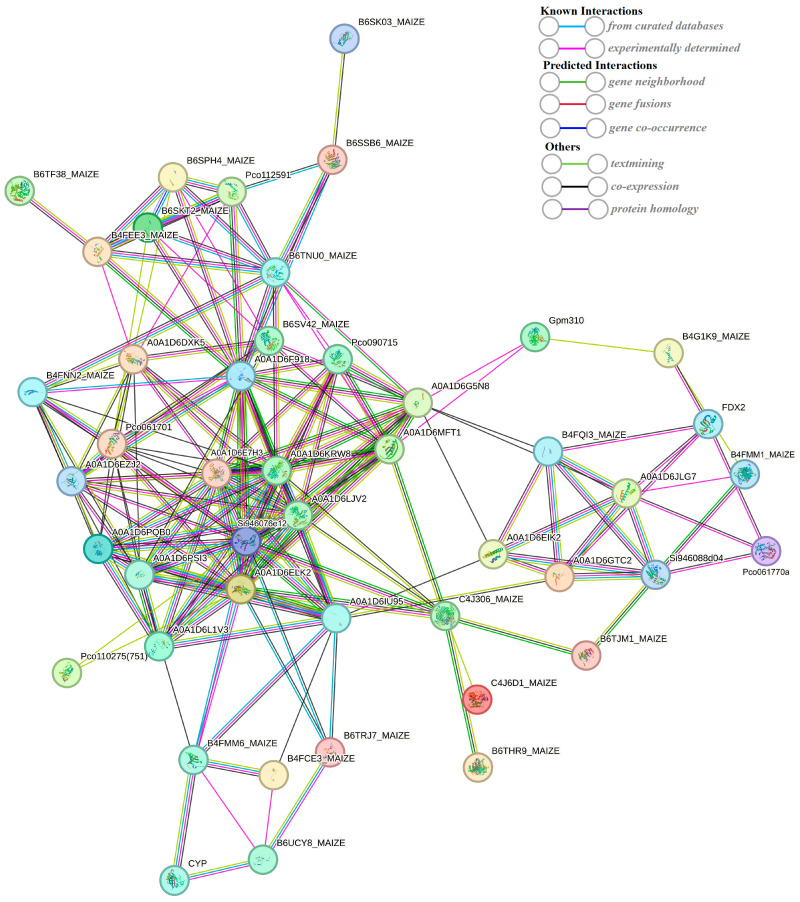
String-based protein–protein interaction (PPI) network.

**Figure 11 plants-13-00065-f011:**
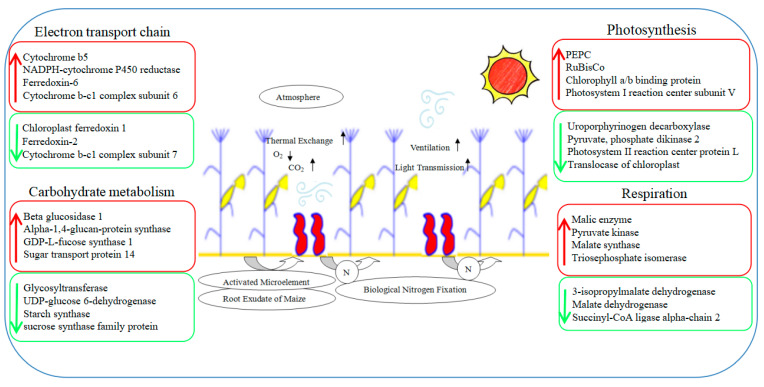
A speculated working model showcases photosynthesis changes in maize for the maize–peanut intercropping system. Red represents up-regulated expression proteins, green represents down-regulated expression proteins.

**Figure 12 plants-13-00065-f012:**
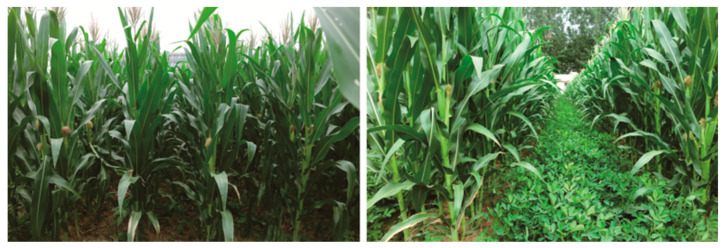
Growth situation in field at skilling stage. Maize monocropping with 60 cm line spacing and 25 cm plant spacing (**left**). Maize–peanut intercropping: row and plant spacing were 40 cm and 50 cm for maize, respectively, and 30 cm and 20 cm for peanut (**right**).

## Data Availability

All data supporting the findings of this study are included in this article.

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
