# Peer review of "Integrative Physiological, Transcriptome, and Proteome Analyses Provide Insights into the Photosynthetic Changes in Maize in a Maize–Peanut Intercropping System"

_plants, 2023, doi:10.3390/plants13010065_

Round 1

Reviewer 1 Report

Comments and Suggestions for Authors

The study presents a comprehensive comparative analysis of the transcriptome and proteome of maize in monocropping and maize-peanut intercropping systems. The data is robust and holds significant practical implications. 

Comments:

1. Fig.7: The font size is small and requires adjustment for improved readability. Please ensure that the text in Figure 7 is legible, and consider increasing the font size as needed.

2. Fig.8: The metabolic pathway mapping in Figure 8 could benefit from a redraw to enhance emphasis and aesthetics. Please revise the figure to ensure clarity and visual appeal while maintaining accuracy.

3. Photosynthetic Efficiency and Biomass Measurement: Throughout the study, the impact of maize and peanut intercropping on maize photosynthetic efficiency has been highlighted. Given that the most direct indicator of improved photosynthetic efficiency is an increase in biomass, it is essential to include measurements of maize biomass in the paper. Please address this omission by providing relevant data or discussing the reasons for its absence.

Comments on the Quality of English Language

Minor English language editing is needed.

Author Response

Dear Editors and Reviewers:

Thank you for your letter and for the reviewers’ comments concerning our manuscript entitled “Integrative physiological, transcriptome and proteome analysis provide insights into the photosynthetic changes of maize in maize-peanut intercropping system” (Manuscript ID: plants-2717693). Those comments are all valuable and very helpful for revising and improving our paper, as well as the important guiding significance to our researches. We have studied comments carefully and have made correction which we hope meet the requirement. Revised portion are marked in red in the paper. The main corrections in the paper and the point-by-point response to the reviewers’ comments are as flowing:

Reviewer #1:

  1. Fig.7: The font size is small and requires adjustment for improved readability. Please ensure that the text in Figure 7 is legible, and consider increasing the font size as needed.

Response: Thank you. We are very grateful for the expert advice. The Figure 7 has been redrawn in order to improve definition.

  1. Fig.8: The metabolic pathway mapping in Figure 8 could benefit from a redraw to enhance emphasis and aesthetics. Please revise the figure to ensure clarity and visualappeal while maintaining accuracy.

Response: Thank you very much for your comment. The Figure 8 has been redrawn in order to improve clarity and visual.

  1. Photosynthetic Efficiency and Biomass Measurement: Throughout the study, the impact of maize and peanut intercropping on maize photosynthetic efficiency has been highlighted. Given that the most direct indicator of improved photosynthetic efficiency is an increase in biomass, it is essential to include measurements of maize biomass in the paper. Please address this omission by providing relevant data or discussing the reasons for its absence.

Response: Thank you. We are very grateful for the expert advice. We have added the biomass of aboveground parts in the Figure 2C.

Special thanks to you for your good comments.

We tried our best to improve the manuscript and made some changes in the manuscript. These changes will not influence the content and framework of the paper. We appreciate for Editors/Reviewers’ warm work earnestly, and hope that the correction will meet with approval.

Once again, thank you very much for your comments and suggestions.

Reviewer 2 Report

Comments and Suggestions for Authors

This is a very meaningful study. But the research methods determine the accuracy of the results. I have two concerns, 1) The author lacks a sampling period, and different sampling periods will lead to differences in results. In addition, how are the materials for sampling selected? 2) The strictness of the screening criteria of the analysis method will also affect the accuracy of the results. Does the author have any basis for the identification criteria of differential proteins in DEPs (absolute value of log2 (1.2-fold change with P value < 0.05)? I feel that the screening criteria It’s not very strict. Could the author please explain why he chose this standard?
Other minor issues:
line 20-22 AQY, LSPn, LSP, LCP, CE, Cisat, Vcmax, Jmax and TPU. Use full name on first occurrence.
Please number the pictures in the order they appear in the text.
line 156 291823, 51015, 22092, 17644 and 4997, the number format needs to be modified
line 169 4997 Number format problem
Figure 7 is not clear and needs to be changed
Figure 8 is not clear and needs to be changed
Figure 11 is not clear and needs to be changed
line 461 HISAT2 and StringTie software Add references
line 477 KEGG software KOBAS 2.0 Add references

Comments on the Quality of English Language

Moderate editing of English language required

Author Response

Dear Editors and Reviewers:

Thank you for your letter and for the reviewers’ comments concerning our manuscript entitled “Integrative physiological, transcriptome and proteome analysis provide insights into the photosynthetic changes of maize in maize-peanut intercropping system” (Manuscript ID: plants-2717693). Those comments are all valuable and very helpful for revising and improving our paper, as well as the important guiding significance to our researches. We have studied comments carefully and have made correction which we hope meet the requirement. Revised portion are marked in red in the paper. The main corrections in the paper and the point-by-point response to the reviewers’ comments are as flowing:

Reviewer #2:

  1. The author lacks a samplingperiod, and different sampling periods will lead to differences in results. In addition, how are the materials for sampling selected?

Response: Thank you. We are very grateful for the expert advice. We have supplemented sampling information in the “Materials and Methods” section.

  1. The strictness of the screening criteria of the analysis method will also affect the accuracy of the results. Does the author have any basis for the identification criteria of differential proteins in DEPs (absolute value of log2 (1.2-fold change with P value < 0.05)? I feel that the screening criteria It’s not very strict. Could the author please explain why he chose this standard?

Response: Thank you very much for your professional comment. The screening criteria of DEPs is very important in proteomic analysis. Before determining the screening criteria, we studied a lot of papers on proteomics. The absolute value of log2 was usually between 1.0~2.0. For example:

“Dwivedi et al., Integrated transcriptome, proteome and metabolome analyses revealed secondary metabolites and auxiliary carbohydrate metabolism augmenting drought tolerance in rice, Plant Physiology and Biochemistry, 201 (2023) 107849”,  Differential   expression (log2 fold change ≥ +1 or ≤ − 1 and FDR ≤ 0.05) of the proteins was performed.

“Zhong et al., iTRAQ analysis of the tobacco leaf proteome reveals that RNA-directed DNA methylation (RdDM) has important roles in defense against  geminivirus-betasatellite infection, Journal of Proteomics 152 (2017) 88”, to be identified as being significantly differentially changed, a protein must pass t-test with p-value < 0.05, with a ratio fold change >1.2 or < 0.83 and be identified in at least two of the three biological replicates.

“Zheng et al., Maize (Zea mays L.) planted at higher density utilizes dynamic light more efficiently, Plant Cell Environ, 2023;1” , In a pairwise comparison of the two planting densities or two growth stages, proteins with a p < 0.05 by a Student's t test and a fold change of >1.30 or <0.77 were considered as differentially abundant proteins (DAPs).

“Yu et al., Proteomics analysis of maize (Zea mays L.) grain based on iTRAQ reveals molecular mechanisms of poor grain filling in inferior grains, Plant Physiology and Biochemistry, 115 (2017) 83”, a change of more than 1.5-fold and P < 0.05 as screening criteria.

“Hola et al., The disadvantages of being a hybrid during drought: A combined analysis of plant morphology, physiology and leaf proteome in maize, PLOS ONE, 12(4): e0176121”, The iTRAQ ratios > 2.0 were considered differentially expressed.

In the present study, the screening criteria (1.2-fold change with P value < 0.05) was suitable, <1.0-fold will result in excessive DEPs, however, > 2.0-fold will result in deficient DEPs.

  1. line 20-22 AQY, LSPn, LSP, LCP, CE, Cisat, Vcmax, Jmax and TPU. Use full name on first occurrence.Please number the pictures in the order they appear in the text.

Response: Thank you very much for your comment. We have used full name on first occurrence and numbered the pictures in the order they appear in the text.

  1. line 156 291823, 51015, 22092, 17644 and 4997, the number format needs to be modified.

Response: Thank you. We are very grateful for the expert advice. We have modified the number format, and unified all the the number format in the manuscript.

  1. line 169 4997 Number format problem.

Response: Thank you. We are very grateful for the expert advice. We have modified the number format.

  1. Figure 7, 8, 11are not clear and needs to be changed.

Response: Thank you very much for your comment. The Figure 7, 8, 11 has been redrawn in order to improve clarity and visual.

  1. line 461 HISAT2 and StringTie software Add references.

Response: Thank you very much for your comment. The suggested references has been added.

  1. line 477 KEGG software KOBAS 2.0 Add references.

Response: Thank you very much for your comment. The suggested references has been added.

Special thanks to you for your good comments.

We tried our best to improve the manuscript and made some changes in the manuscript. These changes will not influence the content and framework of the paper. We appreciate for Editors/Reviewers’ warm work earnestly, and hope that the correction will meet with approval.

Once again, thank you very much for your comments and suggestions.

Reviewer 3 Report

Comments and Suggestions for Authors

This manuscript quantifies the effect of intercropping. 

Major comments:

1. It is not clear the observed effect is due to peanut or added access to sunlight.  A possible control would be the effect of maize at the edge of field or next to an empty strip of field.

2. Figures 2/3/4:  Technical details are missing.  How was each quantities measured (AQY/LSPn, LSP, LCP, Ce, Cisat, Vcmas, Jmas, TPU)?  These methods need to be provided with proper references and methods/equipments.  Additional details like how each error bar was calculated (over how many days, over how many independent measurement) and how p-value was quantified. 

3. Lines 459-462:  What is the reference genome with reference?  Is there a good maize reference genome to use?  If mung bean was used instead, this needs to be justified.  Also, please provide details of RNA-seq processes with references.  I found Figure 6A is striking as so many proteins (20%) were detected without gene expression.

4. Figure 6: To gain the insight of how RNA and protein levels are correlated, it would be great if authors can show the FC of RNA and protein levels as a scatter plot.  This would provide the confidence of transcriptome and proteome analysis.

In addition, it would be helpful if authors can provide Venn Diagram between DEG and DEP.

5. Figure 7: There are too many processes which are essentially the same (e.g. primary metabolic process, organic substance metabolic process, metabolic process).  This makes it very difficult to follow the results.  I suggest using something like GO Slim, so that you can report only the different processes.  In addition, it is difficult to compare DEGs and DEPs in the current format.  I suggest to fix a list of GO/KEGG terms and show p-values for both transcriptome and proteome.

6. Section 2.6 and Fig 10:  I am not sure if this section offers any insights as DEGs by your selection will have different expressions.

Author Response

Dear Editors and Reviewers:

Thank you for your letter and for the reviewers’ comments concerning our manuscript entitled “Integrative physiological, transcriptome and proteome analysis provide insights into the photosynthetic changes of maize in maize-peanut intercropping system” (Manuscript ID: plants-2717693). Those comments are all valuable and very helpful for revising and improving our paper, as well as the important guiding significance to our researches. We have studied comments carefully and have made correction which we hope meet the requirement. Revised portion are marked in red in the paper. The main corrections in the paper and the point-by-point response to the reviewers’ comments are as flowing:

Reviewer #3:

  1. It is not clear the observed effect is due to peanut or added access to sunlight.A possible control would be the effect of maize at the edge of field or next to an empty strip of field.

Response: Thank you very much for your comment. Our early research shows that below ground interspecific interactions stimulate peanut growth, while aboveground interspecific interactions seem to suppress peanut growth but promote maize growth during late coexistence phase. Both aboveground and belowground interspecific interactions had positive effects on advantages of the intercropping, but belowground interaction had more of a contribution than aboveground interaction. It is suggested that belowground interspecific facilitation for peanut and aboveground interspecific competition for maize play the key roles in controlling productivity of a peanut/maize intercropping system. These results were published in “The importance of aboveground and belowground interspecific interactions in determining crop growth and advantages of peanut/maize intercropping, The Crop Journal, 9 (2021) 1460-1469”. In addition, the effects of above-ground and below-ground interspecific interaction on maize/peanut intercropping advantage and photosynthetic characteristics of functional leaves were studied, and contribution of above ground and below-ground to them were analysed by a field experiment with root barrier. The contribution of above-ground interspecific interaction to intercropped peanut were negative effects, below-ground interspecific interaction were positive effects. These results were published in “Effects of root barrier on photosynthetic characteristics and intercropping advantage of maize/peanut intercropping, Plant Physiology Journal, 2016, 52 (6): 886-894”. In the present study, we focus on the mechanism of leaf photosynthesis changes under maize/peanut intercropping system. I hope this explanation could meet with approval.

  1. Figures 2/3/4:  Technical details are missing.  How was each quantities measured (AQY/LSPn, LSP, LCP, Ce, Cisat, Vcmas, Jmas, TPU)?  These methodsneed to be provided with proper references and methods/equipments.  Additional details like how each error bar was calculated (over how many days, over how many independent measurement) and how p-value was quantified. 

Response: Thank you. We are very grateful for the expert advice. We have supplemented methods information in the “Materials and Methods” and “references” section.

  1. Lines 459-462:  What is the reference genome with reference?  Is there a good maize reference genome to use?  If mung bean was used instead, this needs to be justified.  Also, please provide details of RNA-seq processes with references.I found Figure 6A is striking as so many proteins (20%) were detected without gene expression.

Response: We apologize for the negligence. Mung bean is another of our experiment. Sequence alignment was performed with the reference genome of Zea mays (Zm_B73_REFERENCE_NAM_5.0.new). We have made correction in the “Materials and Methods” section. For Figure 6A, thank you very much for your professional questions. We analyzed the possible causes of the “20%”. 1. As a matter of fact, more and more recent studies discovered that many non-coding RNAs have the ability to encode proteins. These non-coding RNAs are not included in the transcriptome. 2. Protein produced by alternative splicing, if the peptide is located in an unknown alternative splicing, it might be out of transcriptome. 3. Translational jumping and translational frameshifting might lead to this situation. We also looked for papers on gene expression, one peper reported molecular mechanisms of cold-tolerance response in japonica rice, 12.6% of proteins were detected without gene expression. “Transcriptome Sequencing and iTRAQ of Different Rice Cultivars Provide Insight into Molecular Mechanisms of Cold-Tolerance Response in Japonica Rice, Rice, (2020) 13:43”. I hope this explanation could meet with approval.

  1. Figure 6: To gain the insight of how RNA and protein levels are correlated, it would be great if authors can show the FC of RNA and protein levels as a scatter plot.This would provide the confidence of transcriptome and proteome analysis.In addition, it would be helpful if authors can provide Venn Diagram between DEG and DEP.

Response: Thank you very much for your valuable comments. We have added the FC of RNA and protein levels as a scatter plot in figure 6D~E. This advice improved our paper very much. Thanks again for you kind help.

  1. Figure 7: There are too many processes which are essentially the same (e.g. primary metabolic process, organic substance metabolic process, metabolic process).  This makes it very difficult to follow the results.I suggest using something like GO Slim, so that you can report only the different processes.In addition, it is difficult to compare DEGs and DEPs in the current format. I suggest to fix a list of GO/KEGG terms and show p-values for both transcriptome and proteome.

Response: Thank you very much for your professional advice. We looked up the information on “GO slims”. GO slims are cut-down versions of the GO ontologies containing a subset of the terms in the whole GO. They give a broad overview of the ontology content without the detail of the specific fine grained terms. GO slims It seems simpler and more efficient. But it is regrettable that we haven't learned yet. In the present study, GO and KEGG enrichment analysis were used to get an overall situation of DEGs and DEPs, in particular, the KEGG analysis included photosynthesis. We believe that the purpose of GO and KEGG enrichment analysis has been achieved. Then, we focused on the DEGs and DEPs in the photosynthetic pathway. I hope this explanation could meet with approval.

  1. Section 2.6 and Fig 10:  I am not sure if this section offers any insights as DEGs by your selection will have different expressions.

Response: Thank you very much for your comment. In most cases, qRT-PCR results are used to verify the reliability of transcriptome. If it's not necessary, section 2.6 and Fig 10 could be deleted.

Special thanks to you for your good comments.

We tried our best to improve the manuscript and made some changes in the manuscript. These changes will not influence the content and framework of the paper. We appreciate for Editors/Reviewers’ warm work earnestly, and hope that the correction will meet with approval.

Once again, thank you very much for your comments and suggestions.

Reviewer 4 Report

Comments and Suggestions for Authors

Overall, the data set is well displayed and explained.  As a fact base research it is supported for an acceptance but Figure 9 may be reserved for lower tone of how the authors speculate the cascade of transcriptome and physiological reactions.

Comments on the Quality of English Language

Minor editorial work may help better legibility.

Author Response

Dear Editors and Reviewers:

Thank you for your letter and for the reviewers’ comments concerning our manuscript entitled “Integrative physiological, transcriptome and proteome analysis provide insights into the photosynthetic changes of maize in maize-peanut intercropping system” (Manuscript ID: plants-2717693). Those comments are all valuable and very helpful for revising and improving our paper, as well as the important guiding significance to our researches. We have studied comments carefully and have made correction which we hope meet the requirement. Revised portion are marked in red in the paper. The main corrections in the paper and the point-by-point response to the reviewers’ comments are as flowing:

Reviewer #4:

  1. Overall, the data set is well displayed and explained.As a fact base research it is supported for an acceptance but Figure 9 may be reserved for lower tone of how the authors speculate the cascade of transcriptome and physiological reactions.

Response: Thank you very much for your comment. Protein is the undertaker in life activities. Figure 9 is based on the changes of differentially expressed proteins (DEPs) involved in photosynthesis efficiency in maize leaves. We used heatmap combined with photosynthetic pathway to display the DEPs in the photosynthetic pathway. This kind of figure is more intuitive and vivid. We hope this explanation could assuage your concerns.

Special thanks to you for your good comments.

We tried our best to improve the manuscript and made some changes in the manuscript. These changes will not influence the content and framework of the paper. We appreciate for Editors/Reviewers’ warm work earnestly, and hope that the correction will meet with approval.

Once again, thank you very much for your comments and suggestions.

Round 2

Reviewer 2 Report

Comments and Suggestions for Authors

The authors have addressed all my comments.

Comments on the Quality of English Language

 Minor editing of English language required

Reviewer 3 Report

Comments and Suggestions for Authors

The manuscript has improved in a number of ways.  The technical details can be still improved.